# Drug or Toxic? A Brief Understanding of the Edible Corolla of *Rhododendron decorum* Franch. by Bai Nationality with Comparative Metabolomics Analysis

**DOI:** 10.3390/metabo14090484

**Published:** 2024-09-04

**Authors:** Weiwei Liu, Ling Wang, Chenghua Yu, Zhongyu Fan, Kaiye Yang, Xinchun Mo

**Affiliations:** 1Lijiang Forest Biodiversity National Observation and Research Station, Kunming Institute of Botany, Chinese Academy of Science, Kunming 650201, China; liuweiwei@mail.kib.ac.cn (W.L.); fanzhongyu@mail.kib.ac.cn (Z.F.); yangkaiye@mail.kib.ac.cn (K.Y.); 2School of Applied Technology, Lijiang Normal University, Lijiang 674199, China; wangling1014@163.com (L.W.); 18468182962@163.com (C.Y.)

**Keywords:** comparative metabolomics analysis, *Rhododendron decorum* Franch., edible corolla, androecium/gynoecium, enrichment analysis of metabolites

## Abstract

*Rhododendron* is a traditional ornamental and medicinal plant in China, renowned for its aesthetic appeal and therapeutic properties. Regarding *Rhododendron decorum* Franch., mainly distributed in Yunnan Province, its corolla is regarded as an edible food by the Bai ethnic group in Yunnan Province. However, it is still unclear why the Bai people choose to use the Rhododendron species in their seasonal diet. Here, we employed comparative metabolomics analysis to explore the variations in the metabolites and the enriched biosynthesis pathways within the different floral organs of *R. decorum* Franch. from Heqing and Yulong County. The metabolite analysis showed that 1340 metabolites were identified from the floral organs in the two regions. Comparing the different flower organs of the same region, 85 differential accumulated metabolites (DAMs) were found from the androecium/gynoecium and corolla in the same region, and 66 DAMs were identified from the same organ in different regions. The KEGG pathway and network analysis revealed significant disparities in both the metabolite composition and enriched pathways among the different floral organs or when comparing the same floral organs across diverse regions, with geographical variations exerting even stronger influences. From the perspective of resource utilization, it was observed that the *R. decorum* Franch. populations in Heqing County exhibited the greater accumulation of secondary metabolites within their flowers, rendering them more advantageous for medicinal purposes, albeit potentially more toxic. This study provides novel insights into the utilization of corollaries for potential de novo pharmacy development.

## 1. Introduction

*Rhododendron* is a collective term for plants belonging to the genus *Rhododendron* in the family Ericaceae, renowned for their significant ecological, ornamental and medicinal value [1]. In China, there are over 600 species of rhododendrons, primarily concentrated in the southwestern region. This area serves as a prominent global distribution center for rhododendrons [2]. The rhododendron contains a wide variety of chemical compounds; within the past 14 years, more than 610 chemical constituents have been identified from rhododendron plants, including flavonoids, terpenes, glycosides, phenols, tannins and volatile oils [3,4,5,6,7]. These components possess medicinal properties and can be used in suppressing coughs, promoting expectoration, alleviating asthma symptoms, reducing blood pressure levels and cholesterol levels and exerting diuretic effects, as well as exhibiting antibacterial activity [8,9,10]. Moreover, they have been demonstrated to show efficacy in the treatment of cardiovascular diseases and rheumatism [8,11,12].

Besides their significant ornamental value, certain rhododendron species are also edible [13,14], particularly among the Bai people who live in the Dali Bai Autonomous Prefecture, Yunnan Province. The Bai ethnic group has a long-standing tradition of consuming the corolla of the rhododendrons, particularly *R. decorum* Franch., which is widely distributed across Yunnan Province, China. This species is not only valued for its ornamental and medicinal properties but also serves as a dietary staple. However, before consumption, the androecium/gynoecium of the rhododendron flower is typically removed, leaving only the intact corolla, for toxin elimination purposes [13]. Moreover, although *R. decorum* Franch. is widely distributed in the northwest of Yunnan Province, it is exclusively harvested and consumed by the Bai people within their residential areas; the neighboring Naxi people do not partake in the collection or consumption of this specific flower.

Metabolomics, which has greatly benefited from advancements in analytical instruments, enables researchers to perform the high-throughput analysis of numerous compounds within plants, with exceptional sensitivity and precision. This advancement has greatly improved the diagnostic capacity of metabolite profiling. Numerous unidentified compounds can be rapidly and effectively discovered; even very small amounts of them can be accurately profiled through this technology [15]. Furthermore, to investigate the spatial distributions of molecules within biological systems, mass spectrometry imaging (MSI) and ultra-high performance liquid chromatography mass spectrometry (UPLC-MS/MS) have emerged as invaluable analytical techniques for the integrated analysis of the constituents within the biological process [16]. Thus, the UPLC-MS/MS technique has been employed to explore metabolic processes across a wide array of plant tissue types, such as the leaves, fruits, stems, roots and seeds, in various model species, including crops and medicinal plants. Moreover, it has been reported that some bioactive constituents with potential health-promoting benefits can be found in onion waste, making it a potentially valuable resource for anti-inflammatory drugs [17]. Comparative metabolomics, a burgeoning discipline that has emerged subsequent to genomics and proteomics, plays an indispensable role in the study of systems biology [18,19]. As a discipline focused on the comprehensive analysis of cellular metabolites at a specific time point, metabolomics has enabled significant strides in fields closely related to human health management, such as pharmaceutical research and development, nutritional food science, toxicology and botany [20,21,22,23,24,25].

Despite the identification of numerous chemicals from rhododendron species, some of which are associated with toxicity, the rationale behind the Bai ethnic group’s consumption of the corolla of *R. decorum* Franch., a specific species distributed near their residences, remains unknown. Several questions remain unanswered: (1) Are there variations in the metabolites between the different floral organs (corolla and androecium/gynoecium) of *R. decorum* Franch.? (2) Do the metabolite profiles differ among distinct geographical distributions of *R. decorum* Franch. within identical floral organs? (3) Which type of variation exhibits greater prominence? Thus, this study aims to elucidate the variations in metabolites among *R. decorum* Franch. from different regions and its diverse floral organs through differential metabolomic analysis. This provides a new perspective for the resource development of this plant species, which possesses both ornamental and medicinal value and potential as a functional food.

## 2. Materials and Methods

### 2.1. Sample Collection

In May 2024, healthy specimens of *R. decorum* Franch. were collected from Heqing County (HQ) (100.075720° E, 26.485620° N, Altitude 3038 m), Dali Bai Autonomous Prefecture, Yunnan Province and Yulong Naxi Autonomous County (LF) (100.180355° E, 27.001036° N, Altitude 3220 m), Lijiang City. Five healthy plants of *R. decorum* Franch. were chosen from the independent populations in the HQ and LF regions. From each plant, three to five intact flowers were collected and subsequently separated into the corolla (HQO and LFO) and androecium/gynoecium (HQI and LFI). After pooling the corolla and androecium/gynoecium samples together, they were divided into three sequencing samples and immediately flash-frozen in liquid nitrogen. The frozen samples (named HQO, LFO, HQI and LFI) were left following metabolite extraction for subsequent metabolite detection with three replicates per sample. All extracted metabolite samples were stored at −80 °C for metabolomics analysis until use.

### 2.2. Metabolite Extraction and Identification by UPLC-ESI-MS/MS

The metabolomics analysis was conducted in collaboration with Biomarker Technologies Co., Ltd. (Beijing, China) and involved a series of specific steps. Initially, 100 mg of crushed, lyophilized powder was suspended in 1200 μL of a 70% methanol aqueous solution and extracted for 180 min at −20 °C. During the extraction process, the mixtures were vortexed once every 30 min, each time lasting 30 s, for a total of 6 times. After centrifugation at 12,000 rpm for 3 min, the supernatant underwent filtration through a 0.22 µm membrane. The quantification of the metabolites in the *R. decorum* Franch. samples utilized ultra-performance liquid chromatography–electrospray ionization tandem mass spectrometry (UPLC-ESI-MS/MS) technology, employing established techniques, using an LC-ESI-MS/MS system (UPLC, gasket-packed UFLC CBM30A, Shimadzu; MS, 6500 QTRAP, Applied Biosystems, Shanghai, China), and analyses were performed on all sample extracts [26,27]. Mobile phases A and B consisted of ultrapure water with formic acid content of 0.1% and acetonitrile with formic acid content of 0.1%, respectively. Following the gradient program, phase B transitioned from 5% to 95% over nine minutes before being immediately lowered back to 5% and adjusted for an additional fourteen minutes after being held for one minute at that level. The column temperature was maintained at 40 °C, while the mobile phase flow rate remained constant at 0.35 mL·min^−1^, and each injection volume was 4 µL.

The ESI process conditions were as follows: the temperature was set at 550 °C; the ion spray voltage (IS) was 5500 V in positive ion mode and −4500 V in negative ion mode; gas I (GSI), gas II (GSII) and the curtain gas (CUR) were adjusted to 50, 60 and 25 psi, respectively; high-collision activated dissociation (CAD) was utilized. QQQ scans were designed as multiple reaction monitoring (MRM) experiments with medium collision gas (nitrogen). The declustering potential (DP) and collision energy (CE) for individual MRM transitions were optimized with additional adjustments. A specific set of MRM transitions was performed for each period based on the eluted metabolites within that timeframe [22,27].

### 2.3. Data Analysis

After normalizing the original peak area information with the total peak area, a follow-up analysis was performed. Principal component analysis (PCA) and Spearman correlation analysis were used to judge the repeatability of the samples within groups and the quality control samples. For the identified compounds, their classification and pathway information was retrieved from the KEGG database, Human Metabolite Database (HMDB) and LIPID MAPS Database [28,29,30]. According to the grouping information, the difference multiples were calculated and compared, and a T test was employed to calculate the differential significance *p*-value of each compound. The R language package ropls was used to perform orthogonal signal correction and partial least squares–discriminant analysis (OPLS-DA) modeling, and 200 permutation tests were performed to verify the reliability of the model [31]. The variable importance in projection (VIP) values of the model were calculated by using multiple cross-validation. The method of combining the difference multiples, the *p*-values and the VIP values of the OPLS-DA model was adopted to screen the differential metabolites. The screening criteria were |log2foldchange| (FC) > 1, *p*-value < 0.05 and VIP value > 1. The differential metabolites with KEGG pathway enrichment significance were calculated using the hypergeometric distribution test [32,33].

## 3. Results

### 3.1. Metabolite Identification and Annotation

Two flower organs of *R. decorum* Franch. from HQ and LF were subjected to untargeted metabolite screening using UPLC-ESI-MS/MS. A total of 1340 metabolites were found in the LFI, LFO, HQI and HQO groups from the two regions; then, the entire metabolites were retrieved and annotated from three databases, namely the KEGG, HMBD and LIPID MAPS databases. The results showed that over one third of them (451 metabolites) were annotated in KEGG, 646 metabolites in HMBD and 200 metabolites were retrieved from LIPID MAPS (Figure 1, Appendix A).

### 3.2. Differential Metabolomic Profiling

#### 3.2.1. Differential Metabolomic Profiling of Floral Organs

A multivariate statistical analysis was carried out to screen the differential metabolites among the different flower organs (corolla and androecium/gynoecium) of *R. decorum* Franch. from HQ and LF (Figure 2). The results of the principal component analysis (PCA) indicated significant differences between the LFI/LFO and HQI/HQO groups, with PC1 accounting for 74.54% in LFI/LFO and 69.31% in HQI/HQO, respectively (Figure 2A,D). The OPLS-DA plots also revealed a significant difference between the different flower organs in the two regions. The interpretation rates for the X and Y matrices are denoted as R2X and R2Y, respectively. The model’s predictive utility is reflected by the value of Q2. A reliable and stable model can be inferred when these three parameters approach one. In this study, the R2X value was 0.888 in the LFI/LFO group and 0.895 in HQI/HQO, the R2Y value was equal to one in the two groups, and the Q2 value was found to be 0.993 for the LFI/LFO group and 0.981 for HQI/HQO (>0.9). These results suggest that the OPLS-DA model adequately explains and predicts the differences between the two groups (Figure 2B,E). The volcano plot provides a more intuitive approach, revealing that, regardless of the distribution area, significant variations in metabolite expression exist among distinct floral organs (corolla and androecium/gynoecium), with a predominant tendency towards downregulation. In total, 1340 DAMs were identified from both LFI/LFO and HQI/HQO. Among the entirety of the identified metabolites, the two groups showed the same metabolite distribution pattern within the corolla and androecium/gynoecium in the same region. Among them, a similarly regulated pattern of metabolites was found in the two regions, where 277 DAMs were upregulated and 540 DAMs were downregulated in the LFI/LFO group, and nearly 40% DAMs (accounting for 523 metabolites) were unchanged. In the HQI/HQO group, 242 DAMs were upregulated, 493 DAMs were downregulated and the rest of the DAMs (605 metabolites) were observed to be unregulated (Figure 2C,F). These results suggest the presence of substantial variations in the metabolites among the floral organs (corolla and androecium/gynoecium).

#### 3.2.2. Differential Metabolomic Profiling in Distinct Geographical Regions

To understand the DAMs between the same floral organs from different regions, we also performed the same analysis within the different geographical regions (Figure 3). The PCA results showed that PC1 had significantly high explanatory power for the model, with explained variances of 65.29% in HQI/LFI and 69.03% in HQO/LFO (Figure 3A,D). The OPLS-DA plots also presented a significant difference within the two groups. The evaluated R2Y value for prediction by the models was equal to one in both groups, and the R2X values exceeded 0.9 (0.986 for HQI/LFI and 0.985 for HQO/LFO) (Figure 3B,E). Among them, the same trend was observed in the comparison between the floral organs in distinct geographical regions. Comparing the androecium/gynoecium in the two regions, 438 DAMs were upregulated in the HQ region, whereas 263 DAMs exhibited downregulation and 639 DAMs were unchanged. Within the corolla, 474 DAMs were upregulated in the HQ region, 260 DAMs exhibited downregulation and 606 DAMs were unchanged. The results showed that the DAMs in the floral organs from the HQ region were expressed more highly than in those in the LF region.

### 3.3. Comparison of Selected Discriminated Metabolites

The absolute FC values (FC (ABS)), *p*-values, VIP scoring and regulation types (either up or down) of the discriminated metabolites identified from the floral organs in the HQ and LF regions are listed in Table 1 and Table 2. In the differential pair comparison, a total of 85 metabolites with FC (ABS) values higher than 10 were significantly altered across the HQI/HQO and LFI/LFO groups, comprising 51 metabolites from the HQI/HQO group and 54 metabolites from the LFI/LFO group. The two groups exhibited identical regulatory patterns, with the notable presence of 20 shared metabolites (Table 1). Most of them were grouped into five categories, such as phenylpropanoids and polyketides (36), lipids and lipid-like molecules (15) and organic oxygen compounds (10). After calculating the regulatory trends, the HQI/HQO group showed an upward trend at the superclass level in alkaloids and their derivatives and organic heterocyclic compounds, while displaying a downward trend in nucleic acids, nucleotides and analogues, and other types showed a mixed trend. Within the LFI/LFO group, only two categories showed significant downregulation, namely organic acids and derivatives and organic oxygen compounds, whereas the other groups displayed a heterogeneous pattern. Moreover, the HQI/HQO group exhibited a significantly higher number of metabolite categories with upregulation/downregulation compared to the LFI/LFO group at the class level (Table 1).

We also compared the FC (ABS) values and *p*-values within the same floral organs from different regions. The results showed that a total of 66 metabolites with an FC (ABS) larger than 10 were identified, comprising 30 metabolites from the LFI/HQI group, 46 metabolites from the LFO/HQO group and only 10 shared metabolites between these two groups (Table 2). Similar to the comparison in the previous analysis, there were three categories with the highest number of differentially expressed metabolites, namely phenylpropanoids and polyketides (30), lipids and lipid-like molecules (46) and organic oxygen compounds (10). After analyzing the regulatory patterns, it was observed that the HQI/LFI group exhibited an increasing trend in organic acids and derivatives at the superclass level. Conversely, a decreasing trend was observed in alkaloids and derivatives and organic oxygen compounds. Additionally, other categories displayed a combined pattern. In the HQO/LFO group, only benzenoids and organic nitrogen compounds exhibited upregulation, and two categories showed downregulation, namely alkaloids and derivatives and organoheterocyclic compounds. Meanwhile, the others displayed a heterogeneous pattern. Interestingly, the HQI/LFI group and the HQO/LFO group showed significantly differential regulation patterns at the class level, excluding the categories of indoles and derivatives and tannins. The results showed that the significant differences within the HQI/LFI and HQO/LFO groups encompassed variations in both metabolites and regulatory patterns.

### 3.4. Functional Annotation and Enrichment Analysis of Differential Metabolites

#### 3.4.1. Classification of Differential Metabolite Pathways

To elucidate the metabolic pathways responsible for producing these distinct metabolites, KEGG pathway analysis was employed to the annotate the DAMs. The top 20 pathways with the highest number of DAMs were selected and are shown in the figures. There were no significant differences observed between the LFI/LFO group and the HQI/HQO group at level 1 of the KEGG pathway (Figure 4). However, at level 2 of the KEGG pathway, a distinct arginine and proline metabolism pathway was exclusively identified in the LFI/LFO group within amino acid metabolism. These results suggested that the *R. decorum* Franch. flowering in the LF region needed more proteins for biosynthesis than that of the HQ region. Additionally, in the biosynthesis of other secondary metabolites, the LFI/LFO group exhibited a specific glucosinolate biosynthesis pathway, while the HQI/HQO group demonstrated a broader biosynthesis pathway encompassing various plant secondary metabolites. Furthermore, in the carbohydrate metabolism pathway, only the HQI/HQO group displayed a unique starch and sucrose metabolite pathway, suggesting that the flower might accumulate more saccharides in the HQ region.

Interestingly, in the comparison within the HQI/LFI and HQO/LFO groups, significant distinctions were observed at level 1 of the KEGG pathway (Figure 5). The metabolism of cofactors and vitamins pathway was uniquely exhibited in the HQI/LFI group, while the lipid metabolism pathway was specifically enriched in the HQO/LFO group. At level 2 of the KEGG pathway, the HQI/LFI group demonstrated enrichment in pathways related to arginine and proline metabolism and the biosynthesis of various plant secondary metabolites, as well as porphyrin metabolism within the cofactors and vitamins metabolic pathway. Conversely, the galactose metabolism pathways, pyrimidine metabolism pathways and glycerophospholipid within lipid metabolism were specifically enriched in the HQO/LFO group (Figure 5).

#### 3.4.2. Enrichment Analysis of Differential Metabolites

To enhance the understanding of the distinct metabolites in the biosynthesis pathway, we functionally annotated and performed an enrichment analysis of the differentially abundant metabolites by comparing them against the KEGG database, including LFI/LFO, HQI/HQO, HQI/LFI and HQO/LFO. The top 20 pathways that possessed the highest number of differentially expressed metabolites in each group are shown with the KEGG enrichment diagram for these specific metabolites in the figures.

In the LFI/LFO group, 221 metabolites were annotated into 81 KEGG pathways. Of these, 129 DAMs were found (Appendix A). The enrichment analysis revealed that significant differences were found, with more enriched factors in the flavonoid biosynthesis pathway and purine metabolism pathway. The observed differences in these pathways were further verified with robust statistical significance, evidenced by the fact that they possessed the smallest *p*-values. Both upregulated and downregulated differential metabolites could be found in these pathways. In contrast, other pathways had larger *p*-values, suggesting that they had lower reliability in terms of significance (Figure 6A).

In the HQI/HQO group, the annotation of the metabolites was the same as in the LFI/LFO group; 221 metabolites were mapped into 86 KEGG pathways and 143 DAMs were found to be differential metabolites (Appendix A). Further enrichment results showed that the enrichment pathways were the aminoacyl-tRNA biosynthesis pathway and glucosinolate biosynthesis pathway. The D-amino acid metabolism pathway exhibited a mixed regulation pattern, with both upregulated and downregulated mechanisms observed within the pathway. Meanwhile, the enrichment of many differential metabolites was involved in the flavonol biosynthesis pathway, purine metabolism pathway and flavone and flavonol biosynthesis pathway. However, they possessed higher *p*-values and showed relatively lower significance compared to the pathways in the LFI/LFO group (Figure 6B).

After performing the comparative analysis of the differential metabolites enriched between the LFI/LFO group and the HQI/HQO group, we observed significant differences in the enriched metabolic pathways within the different floral organs (corolla, androecium/gynoecium) of *R. decorum* Franch. However, only a few reliable pathways showed significant enrichment among these enriched pathways, and, within these enriched pathways, the differential metabolites exhibited both upregulation and downregulation patterns. The results indicated that there were different metabolite biosynthesis mechanisms within the different floral organs (corolla, androecium/gynoecium) of *R. decorum* Franch. in the same region.

In the HQI/LFI group, we also identified 221 metabolites within this group, where 122 differential metabolites were found. The entirety of the identified metabolites were classified into 86 KEGG pathways (Appendix A). Among the identical pathways, the multiple significantly enriched pathways were observed to be varied, including aminoacyl-tRNA biosynthesis; glycine, serine and threonine metabolism; D-amino acid metabolism; arginine biosynthesis; and flavonoid metabolism. Furthermore, all enriched pathways exhibited a high level of statistical significance. Notably, both the glycine, serine and threonine metabolism pathway and the flavonoid metabolism pathway were found to exhibit upregulation for all differentially expressed metabolites (Figure 7A).

The HQO/LFO group exhibited a comparable pattern to the HQI/LFI group, with 221 metabolites classified into 87 KEGG pathways. A total of 132 differential metabolites were also identified (Appendix A), and several metabolic pathways displayed significantly enriched differential metabolites. These pathways encompassed aminoacyl-tRNA biosynthesis; pyrimidine metabolism; cyanoamino acid metabolism; glycine, serine and threonine metabolism; and glucosinolate biosynthesis. Moreover, these pathways demonstrated significantly enhanced enrichment significance. It is noteworthy that all differential metabolites in these pathways, except for purine metabolism, exhibited upregulation (Figure 7B).

#### 3.4.3. Network Analysis for Enrichment of DAMs

To verify whether the DAMs were correlated within different pathways, all DAMs were subjected to a network analysis. The results revealed that twelve DAMs exhibited significant regulation within different pathways (Table 3). In total, 37 DAMs were found to have correlations in five metabolism pathways within the HQI/HQO group, namely aminoacyl-tRNA biosynthesis (ko00970), D-amino acid metabolism (ko00470), flavone and flavonol biosynthesis (ko00944), glucosinolate biosynthesis (ko00966) and purine metabolism (ko00230) (Figure 8). Of these, five metabolites were found to be significantly regulated within different pathways, where S-(4-methylthiobutylthiohydroximoyl)-L-cysteine (NEG_t83) was upregulated in the glucosinolate biosynthesis pathway and apiin (POS_q83) was also upregulated in the flavone and flavonol biosynthesis pathway. 5-Hydroxyisourate (POS_t44), myricetin (NEG_q313) and L-arginine (POS_q213) were found to be downregulated within the purine metabolism pathway, flavone and flavonol biosynthesis pathway and glucosinolate biosynthesis pathway, respectively. Fourteen metabolites were also found to be shared within three metabolism pathways, and there was no metabolite that had a correlation with other pathways in the flavone and flavonol biosynthesis pathway (Figure 8A).

In the LFI/LFO group, 50 DAMs were annotated in five metabolism pathways, namely flavonoid biosynthesis (ko00941), D-amino acid metabolism (ko00470), isoflavone biosynthesis (ko00943), sulfur metabolism (ko00920) and purine metabolism (ko00230). Five metabolites exhibited significant downregulation in the flavonoid biosynthesis pathway, namely tricetin (NEG_q424), (+)-gallocatechin (NEG_q207), (-)-epigallocatechin (POS_q148), dihydromyricetin (NEG_q160) and myricetin (NEG_q313) (Figure 8B). The chemical apigenin (NEG_93) formed a connection between flavonoid biosynthesis and isoflavone biosynthesis, whereas the other pathways were shared with three metabolites, namely L-glutamine (POS_q221), L-serine (NEG_q277) and sulfate (NEG_q407). The downregulation of flavonoid biosynthesis in both the LFI and LFO groups suggests that the corolla exhibits the more pronounced synthesis of flavonoids, as propagative organs still need to prioritize various reproductive functions. It is plausible that the synthesis of flavonoids in the corolla serves as a defense mechanism against insects or climate factors.

Here, we also compared the different floral organs within the HQ and LF regions. In the HQI/HQO group, the diversity of the typical differential metabolites and metabolic pathways indicates the higher complexity of the metabolic characteristics among different the flower organs in Heqing County. In contrast, in the LFI/LFO group, only flavonoids were identified as unique metabolites, and flavonoid biosynthesis was the most significant metabolic pathway. This indicates that the main metabolite composition difference between the different flower organs in Yulong County lies in flavonoids.

Moreover, to examine the same organs in different regions, another network analysis was conducted. In total, 36 DAMs were mapped into five metabolism pathways, namely aminoacyl-tRNA biosynthesis (ko00970), D-amino acid metabolism (ko00470), flavone and flavonol biosynthesis (ko00944), arginine biosynthesis (ko00220) and glycine, serine and threonine metabolism (ko00260) (Figure 9). In the HQI/LFI group, (−)-epigallocatechin (POS_q148) emerged as the most significantly differentially expressed metabolite, exhibiting upregulation and belonging to the flavonoid biosynthesis pathway. Additionally, it actively participates in the biosynthesis of secondary metabolites pathway (Figure 9A, Table 3). Uridine 5′-monophosphate emerged as the most significantly differentially expressed metabolite in the HQO/LFO group. In addition to its involvement in the pyrimidine metabolism pathway, it plays a pivotal role in various other metabolic pathways, including metabolic pathways, nucleotide metabolism and the biosynthesis of cofactors (Figure 9B, Table 3). The results indicate that there are significant variations in the metabolites and metabolic pathways of the same floral organs across different regions.

## 4. Discussion

Rhododendrons represent an important type of historical heritage in China, serving both ornamental and medicinal purposes. The toxicity of rhododendrons has also been extensively documented in the Compendium of Materia Medica, authored by Li Shizhen during the Ming Dynasty [7,34]. *R. decorum* Franch., a widely distributed rhododendron species with significant medicinal value in the northwestern region of Yunnan Province, China, is also harvested and utilized by the local Bai community [13]. Considering that only the corolla of *R. decorum* Franch. is locally consumed and there is minimal collection from other regions, we conducted a metabolomics analysis to investigate the variations in metabolites across different regions and floral organs.

In this study, we employed a differential metabolomics analysis at two levels, distinct floral organs and diverse regions, based on 1340 identified metabolites. The findings revealed significant disparities in the metabolite profiles at both hierarchical tiers. Furthermore, significant variations in metabolites were observed between different floral organs within the HQI/HQO group and the LFI/LFO group, indicating substantial intergroup differences in differential compounds. These disparities persisted even among classes with similar regulatory patterns, as distinct discrepancies in significantly altered metabolites were identified, thus confirming the physiological functional disparities between different floral organs [35,36]. For instance, a comparison between the HQI/LFI and HQO/LFO groups revealed that prenol lipids exhibited upregulation in the HQO/LFO group, whereas downregulation was observed in the HQI/LFI group. The accumulation of prenol lipids indicates their involvement in polysaccharide biosynthesis and suggests the higher edible value of the corolla compared to the androecium/gynoecium, with a significant advantage for the HQO group over the LFO group [37,38]. With respect to flavonoids, although both the HQI/LFI and HQO/LFO groups exhibited the upregulation of these metabolites, these upregulated compounds were completely distinct from each other. This suggests that, even within the same floral organ in different regions, there was variation in the accumulation of flavonoid metabolites. Furthermore, flavonoids were proven to possess diverse biological qualities, such as antioxidant properties and potential as a treatment for cardiovascular diseases. Additionally, they possess toxicity, implying that regional influences on differential metabolite formation might outweigh those arising from variations in the floral organs [39,40,41,42,43].

By performing a functional annotation and enrichment analysis of the identified differential metabolites, we identified discrepancies between the LFI/LFO group and the HQI/HQO group at the secondary level of KO categories. In terms of biosynthetic pathways for secondary metabolites, both groups exhibited the higher abundance of differential metabolites in the flavonoid biosynthesis pathway, indicating that the androecium/gynoecium could accumulate a greater amount of flavonoids compared to the corolla. This indirectly supports the practice of removing the androecium/gynoecium when consuming rhododendron flowers [13]. Additionally, variations in metabolic pathways were observed at the primary level of KO classification between the HQI/LFI group and HQO/LFO group. These differences were particularly evident in the arginine and proline metabolism pathways, as well as the porphyrin metabolism pathway, which play crucial roles in the accumulation of plant secondary metabolites [44]. In terms of the corolla, the distinct galactose metabolism pathways and glycerophospholipid metabolism pathways elucidate the significant process of sugar accumulation specifically within corolla tissue, with a greater cumulative effect observed in the HQO group compared to the LFO group.

Furthermore, the presence of enriched pathways related to flavonoid biosynthesis and purine metabolism within the LFI/LFO group indicates the heightened accumulation of secondary metabolites specifically within the reproductive organs (androecium/gynoecium) as opposed to the petals (corolla) among individuals from the Yulong County population located within Lijiang City. Moreover, in addition to the aforementioned identified pathways, there was also significant enrichment observed within the HQI/HQO group with respect to the aminoacyl-tRNA biosynthesis pathway, D-amino acid metabolism pathway and glucosinolate biosynthesis pathway. The former two pathways are involved in active protein synthesis, while glucosinolate biosynthesis serves as a model system for the investigation of secondary metabolites and plays an important role in plant defense and sulfur element regulation [45,46,47]. Thus, *R. decorum* Franch. in Heging County, relative to Yulong County in Lijiang City, exhibited a more pronounced accumulation pattern of secondary metabolites in the androecium/gynoecium compared to the corolla. This observation suggests that the androecium/gynoecium of this population might possess its own defensive matrix, further reinforcing the necessity for local residents to remove the androecium/gynoecium when consuming the flowers.

Comparing the different regions, the HQI/LFI group exhibited significant enrichment effects in various metabolic pathways, including aminoacyl-tRNA biosynthesis; glycine, serine and threonine metabolism; D-amino acid metabolism; arginine biosynthesis; and flavonoid metabolism. Further analysis revealed that there was more pronounced activity of protein synthesis and the accumulation of secondary metabolites. Moreover, the upregulation of glycine, serine, and threonine metabolism, as well as flavonoid metabolism, indicated that the androecium/gynoecium of this population possessed higher medicinal value or greater toxicity [48,49]. Regarding the HQO/LFO group, significant enrichment effects were observed in aminoacyl-tRNA biosynthesis; pyrimidine metabolism; cyanoamino acid metabolism; glycine, serine and threonine metabolism; glucosinolate biosynthesis; and glycerophospholipid metabolism. This suggests that, in the corolla, the HQO group accumulated a larger amount of secondary metabolites and carbohydrates, with the more pronounced accumulation of secondary metabolites [50].

By integrating the results of the differential metabolic pathway analysis from both the HQI/LFI group and the HQO/LFO group, it can be inferred that the androecium/gynoecium of *R. decorum* Franch. exhibits enhanced medicinal properties (or toxicity) in comparison to its corolla. The population in Heqing County, Dali Prefecture demonstrates enhanced medicinal properties (or toxicity) compared to the population in Yulong County, regardless of whether it pertains to the corolla or androecium/gynoecium. This suggests the presence of plant defense mechanisms within this particular population [51]. The enrichment performance of the *R. decorum* Franch. metabolites in Heqing County, Dali Prefecture suggests that it might initially have served as a traditional folk medicine among the local Bai ethnic group, with its edibility emerging later on [9,13,52].

After thoroughly examining the comprehensive results of the enrichment analysis conducted on the differential metabolites, it becomes evident that the enrichment effect of these metabolites is significantly stronger when comparing different regions with the same floral organs, as opposed to comparing the same region with different floral organs. This observation is supported not only by the greater number of enriched pathways and higher confidence levels for the differential metabolites between different regions with the same floral organs, but also by the more consistent and uniform regulatory pattern exhibited across all enriched pathways. The impact of geographical (environmental) factors on the enrichment effects of plant differential metabolites warrants further attention, thereby highlighting the existence and significance of endemism (regional particularity) in resource plants during their utilization process [53].

The KEGG enrichment network analysis allowed for the clear identification of the key differential metabolites in various enriched pathways. Within the HQI/HQO group, flavonoid metabolism involved apiin and myricetin biosynthesis, both of which exhibit broad activity encompassing potent antioxidant, anticancer, antidiabetic and anti-inflammatory effects [54,55,56]. Similarly, the primary differential compounds within the LFI/LFO group also participated in flavonoid metabolism. In contrast, within the HQI/LFI group, (−)-epigallocatechin stood out as the predominant differentially expressed compound. It is the key component of green tea catechins derived from tea leaves and has demonstrated exceptionally strong antioxidant properties, exceeding those of vitamin C by a minimum of 100 times and vitamin E by 25 times. It also has been proven to protect cells and DNA from damage associated with cancer, cardiovascular diseases and other major illnesses, which can be attributed to its potential for antioxidant and free radical activity [57,58]. Moreover, it has exhibited the ability to lower blood lipids and glucose levels while concurrently preventing cardiovascular diseases [59,60]. Interestingly, the primary differential compound in the HQO/LFO group was found to be uridine 5′-monophosphate, contrasting with the other compounds. As a degradation product of nucleotides and nucleic acids, it has been scientifically proven to possess exceptional energy stability characteristics and exhibit antiarrhythmic effects, thus demonstrating high potential for drug development [61]. To conclude, we have strong evidence to suggest that *R. decorum* Franch. possesses certain medicinal value and holds potential for development. However, when considering its utilization as a natural health food, caution should be exercised due to its potential toxic effects [13,62]. Moreover, we can investigate the gene regulation mechanism through multi-omics analysis for further research. The significant findings of this study hold immense value in aiding the development of rhododendron products.

## 5. Conclusions

*R. decorum* Franch., as a valuable botanical resource, possesses both medicinal and edible flowers. In this study, to understand the consumption of the corolla of *R. decorum* Franch. by the Bai population, we collected the floral organs from the HQ and LF regions and detected the metabolites generated by the corolla and androecium/gynoecium. In total, 1340 metabolites were found in the four floral organs derived from the HQ and LF regions. Among them, 85 DAMs were found to be regulated in the different floral organs, where 12 DAMs were significantly regulated within the different groups. The results suggested that significantly divergent metabolite biosynthesis profiles and accumulation levels existed among the different floral organs, with the androecium/gynoecium exhibiting higher secondary metabolite accumulation and more potent medicinal effects (toxicity) compared to the corolla. This is the reason that the Bai people choose to consume the corolla of *R. decorum* Franch. and remove the other floral organs. Furthermore, even within populations inhabiting distinct regions, substantial variations in the metabolite composition and accumulation levels are observed for the same floral organ. The notion of plant endemism has been further substantiated through the differential accumulation of secondary metabolites. The medicinal properties (toxicity) of *R. decorum* Franch. flowers may vary due to population variations. It is not advisable for individuals to handle and consume the flowers of *Rhododendron* privately without undergoing professional processing procedures that adhere to standardized protocols.

## Figures and Tables

**Figure 1 metabolites-14-00484-f001:**
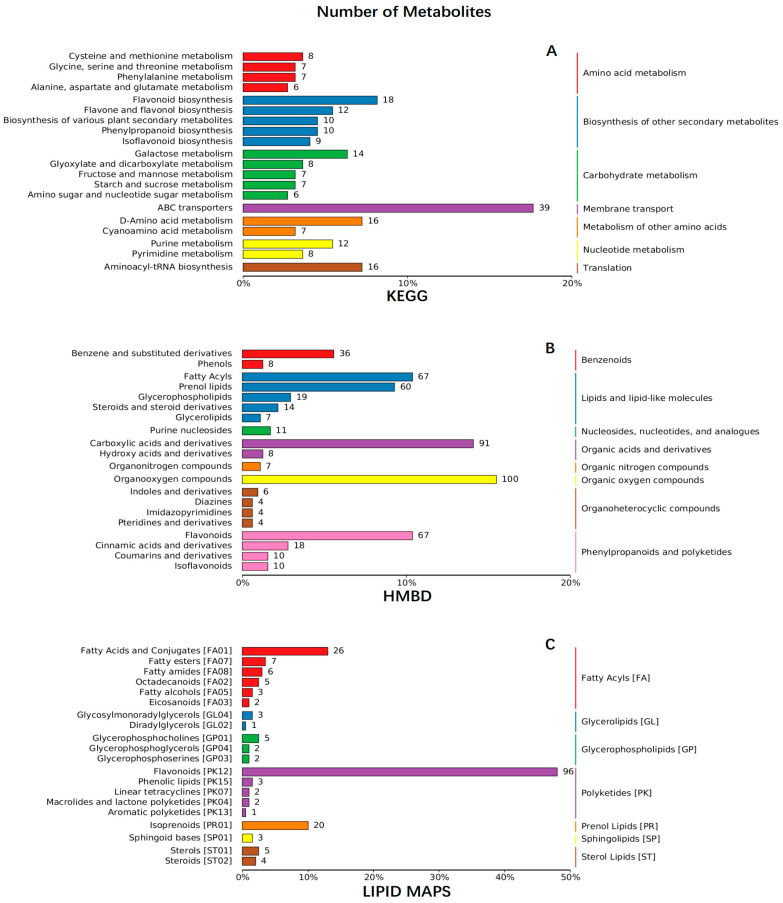
The KEGG, HMBD and LIPID MAPS annotations of the entire metabolites identified from the floral organs of *R. decorum* Franch., collected from HQ and LF. (**A**) KEGG annotation of metabolites in HQ and LF. (**B**) HMBD annotation of metabolites in HQ and LF. (**C**) LIPID MAPS annotation of metabolites in HQ and LF. The colors in the three figures represent different pathways or chemicals. The number at the tail of each line represents the number of metabolites identified from the floral organs of *R. decorum* Franch. collected from HQ and LF.

**Figure 2 metabolites-14-00484-f002:**
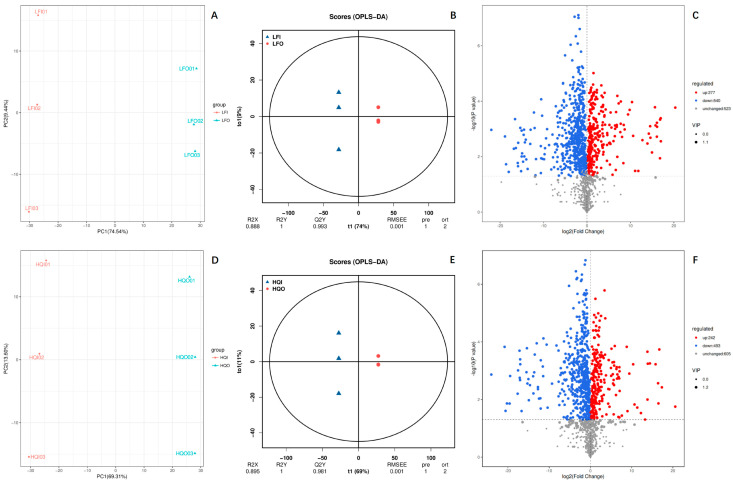
The multivariate statistical analysis of the identified DAMs in LFI/LFO and HQI/HQO of *R. decorum* Franch. (**A**) PCA plot of DAMs within LFI/LFO group, (**B**) OPLS−DA plot of DAMs within LFI/LFO group, (**C**) volcano plot of DAMs within LFI/LFO group, (**D**) PCA plot of DAMs within HQI/HQO group, (**E**) OPLS−DA plot of DAMs within HQI/HQO group, (**F**) volcano plot of DAMs within HQI/HQO group.

**Figure 3 metabolites-14-00484-f003:**
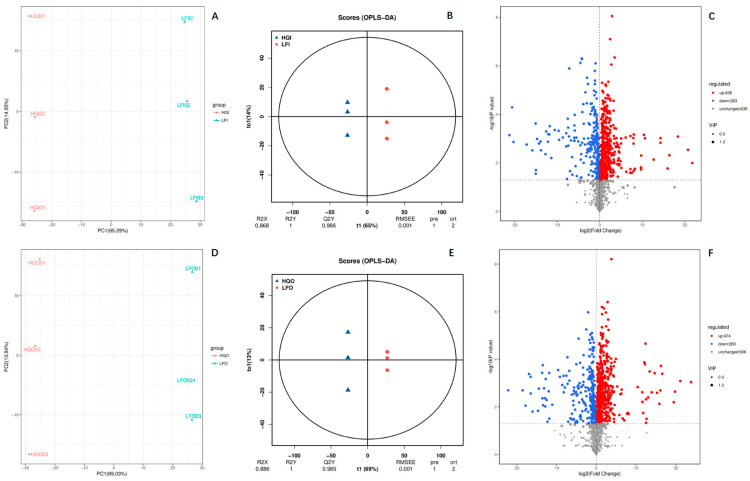
The multivariate statistical analysis of the identified DAMs in HQI/LFI and HQO/LFO of *R. decorum* Franch. (**A**) PCA plot of DAMs within HQI/LFI group, (**B**) OPLS−DA plot of DAMs within HQI/LFI group, (**C**) volcano plot of DAMs within HQI/LFI group, (**D**) PCA plot of DAMs within HQO/LFO group, (**E**) OPLS−DA plot of DAMs within HQO/LFO group, (**F**) volcano plot of DAMs within HQO/LFO group.

**Figure 4 metabolites-14-00484-f004:**
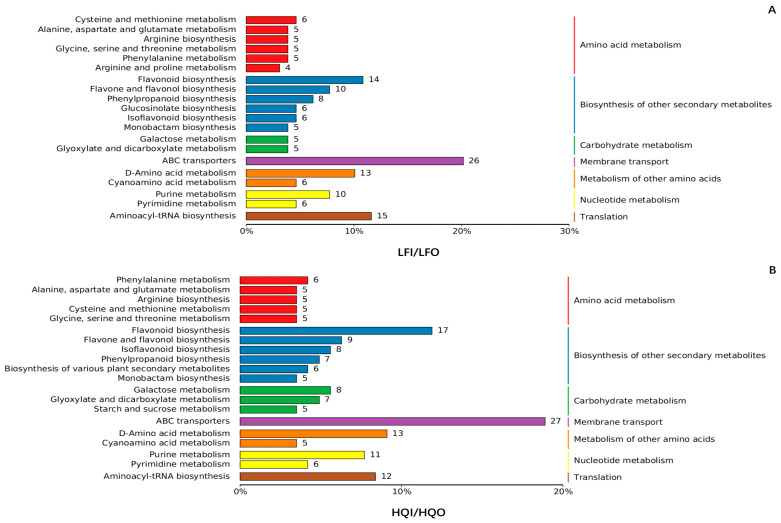
The KEGG pathway analysis of the differential metabolites within the LFI/LFO group and HQI/HQO group of *R. decorum* Franch. (**A**) KEGG pathway analysis of the differential metabolites in LFI/LFO group. (**B**) KEGG pathway analysis of the differential metabolites in HQI/HQO group. The differently colored bars represent hierarchical annotations of KEGG pathways, corresponding to KO categories in level 2 and KEGG pathway names. The length of each bar accurately reflects the number of differential metabolites annotated within that specific pathway.

**Figure 5 metabolites-14-00484-f005:**
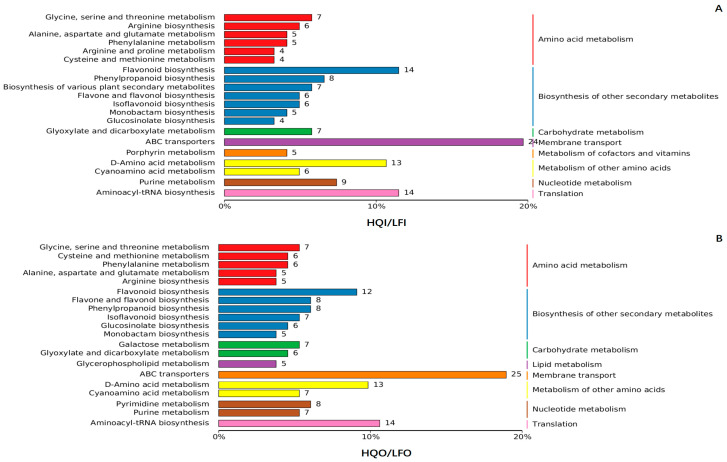
The KEGG pathway analysis of the differential metabolites within the HQI/LFI group and HQO/LFO group of *R. decorum* Franch. (**A**) KEGG pathway analysis of the differential metabolites in HQI/LFI group. (**B**) KEGG pathway analysis of the differential metabolites in HQO/LFO group. The differently colored bars represent hierarchical annotations of KEGG pathways, corresponding to KO categories in level 2 and KEGG pathway names. The length of each bar accurately reflects the number of differential metabolites annotated within that specific pathway.

**Figure 6 metabolites-14-00484-f006:**
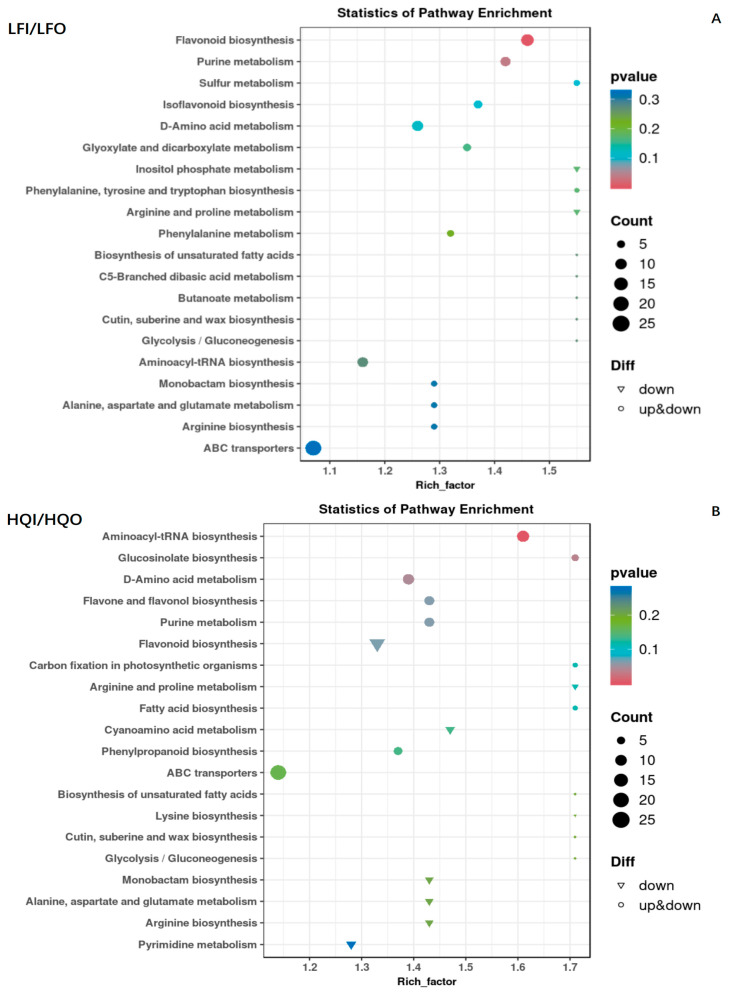
Scatter diagram of the enrichment of differential metabolites in the KEGG pathways within the LFI/LFO group and HQI/HQO group. (**A**) KEGG enrichment analysis of differential metabolites in LFI/LFO group. (**B**) KEGG enrichment analysis of differential metabolites in HQI/HQO group. The ordinate is the name of the KEGG metabolic pathway, and the abscissa is the enrichment factor. The dot size represents the number of metabolites annotated, the triangle denotes downregulation and the circle represents mixed regulation.

**Figure 7 metabolites-14-00484-f007:**
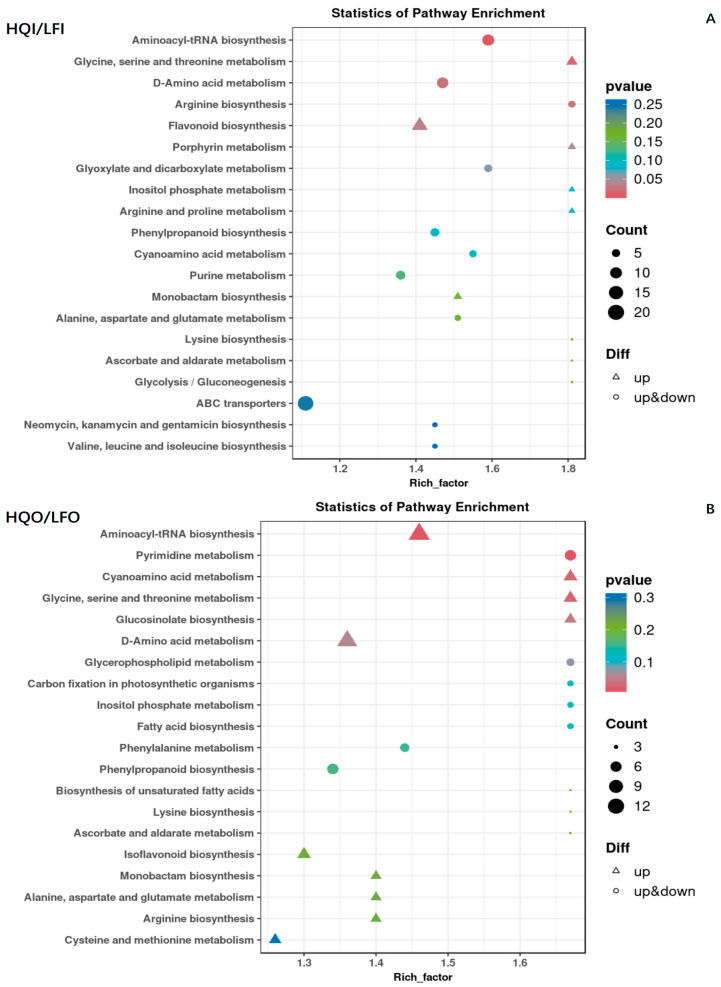
Scatter diagram of the enrichment of differential metabolites in the KEGG pathways within the HQI/LFI group and HQO/LFO group. (**A**) KEGG enrichment analysis of differential metabolites in HQI/LFI group. (**B**) KEGG enrichment analysis of differential metabolites in HQO/HQO group. The ordinate is the name of the KEGG metabolic pathway, and the abscissa is the enrichment factor. The dot size represents the number of metabolites annotated, the triangle denotes downregulation and the circle represents mixed regulation.

**Figure 8 metabolites-14-00484-f008:**
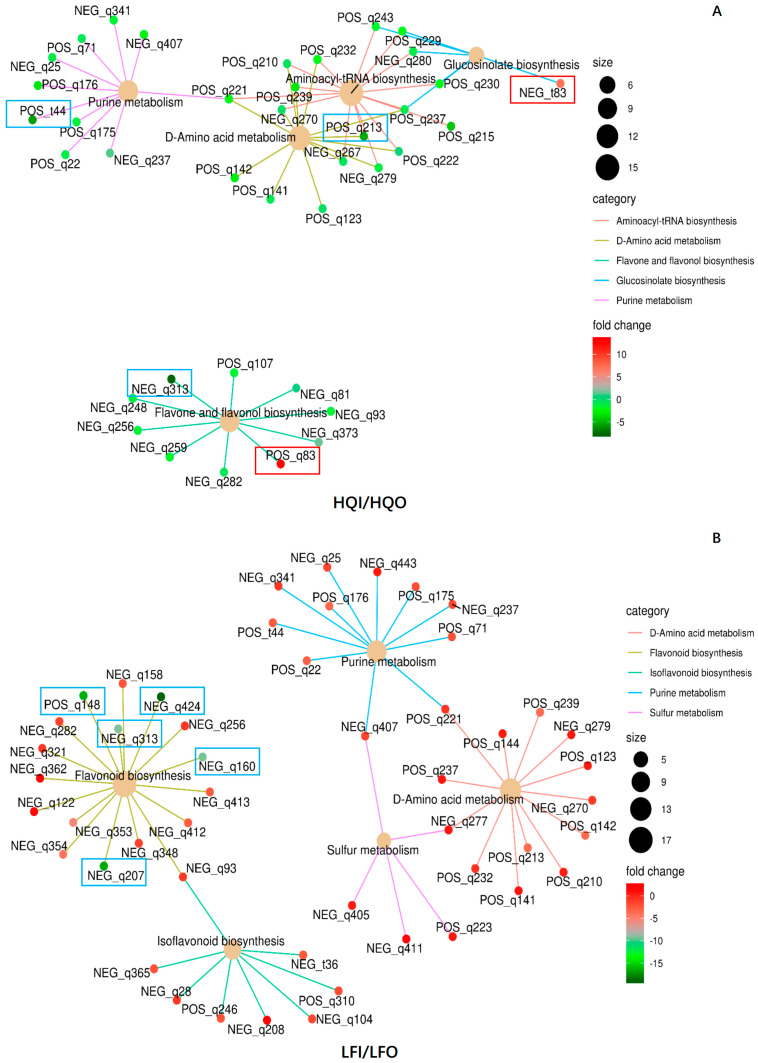
The network diagram of the significant pathways in the HQI/HQO group and LFI/LFO group. (**A**) Network analysis of the significant pathways in HQI/HQO group. (**B**) Network analysis of the significant pathways in LFI/LFO group. Different categories are represented by different colors, and the dot size indicates the number of metabolites involved. The colors of the dots represent the FC values. The metabolite ID in the box denotes the significantly regulated ones, where the blue color indicates that it is downregulated and red indicates that it is upregulated.

**Figure 9 metabolites-14-00484-f009:**
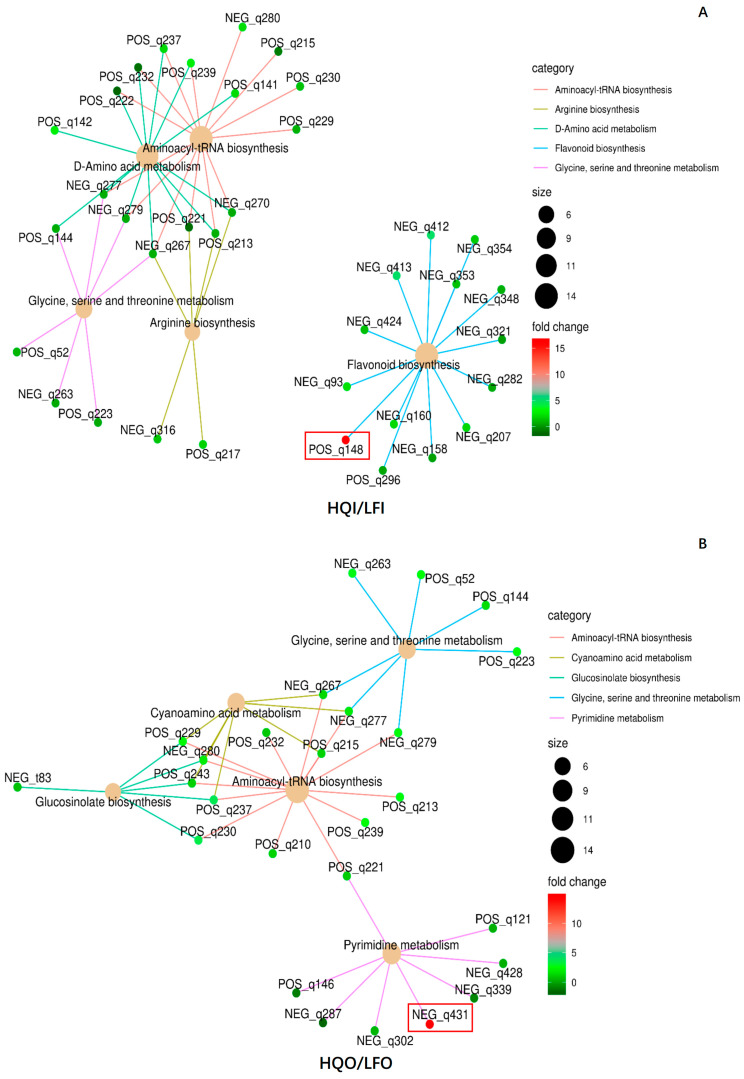
The network diagram of the significant pathways in the HQI/LFI group and HQO/LFO group. (**A**) Network analysis of the significant pathways in HQI/LFI group. (**B**) Network analysis of the significant pathways in HQO/LFO group. Different categories are represented by different colors, and the dot size indicates the number of metabolites involved. The colors of the dots represent the FC values. The metabolite ID in the red box denotes the significantly upregulated ones.

**Table 1 metabolites-14-00484-t001:** Comparative regulation (up/down) of the mutually shared and unique metabolites between the HQI/HQO and LFI/LFO groups.

Superclass	Class (HMDB_Taxonomy)	Metabolite	HQI/HQO				LFI/LFO			
			FC (ABS)	*p*-Value	VIP	Regulated	FC (ABS)	*p*-Value	VIP	Regulated
Alkaloids and derivatives (3)		Riddelline	13.84	0.01	1.18	up	16.09	0.00	1.15	up
		Acetyl-*L*-Carnitine (Hydrochloride)					13.92	0.00	1.15	up
		Salsolidine					14.72	0.02	1.12	down
Benzenoids (7)	Benzene and substituted derivatives	Ethyl Pivaloylacetate	17.06	0.00	1.20	down	14.58	0.00	1.16	down
		Dibenzyl Disulfide	17.55	0.00	1.20	down				
		2,6-Dihydroxybenzoic Acid	17.04	0.00	1.20	down				
		4-Acetoxy-3,5-Dimethoxybenzoic Acid					11.08	0.03	1.09	up
		Ethyl 2,4,6-Trihydroxybenzoate					10.62	0.04	1.07	down
	Naphthalenes	Apiin	13.26	0.05	1.09	up				
	Phenols	Syringylpropane					15.94	0.02	1.12	down
Lipids and lipid-like molecules (15)	Fatty acyls	2,6,4′-Trihydroxy-4-Methoxybenzophenone	15.80	0.00	1.20	up	16.97	0.00	1.16	up
		9(*S*)-Hotre	13.03	0.00	1.20	down	12.98	0.00	1.16	down
		Suberic Acid					20.33	0.00	1.16	up
		Dodecanedioic Acid					15.37	0.00	1.16	down
	Glycerophospholipids	(2*S*,3*R*,4*S*)-4-Hydroxyisoleucine	10.37	0.00	1.20	up				
	Prenol lipids	Menaquinone-4	15.40	0.00	1.20	down	15.06	0.00	1.16	down
		Isopulegol	14.93	0.01	1.18	down	15.84	0.00	1.15	down
		Shanzhiside	13.50	0.00	1.20	down	10.54	0.01	1.13	down
		Monotropein	20.81	0.00	1.20	down				
		Carvacrol	15.04	0.00	1.20	down				
		Oleoside	11.40	0.00	1.20	down				
		(*R*)-(+)-Citronellal					16.13	0.00	1.16	up
		Geniposidic Acid					11.86	0.05	1.06	down
	Steroids and steroid derivatives	2-Phenylethylamine (Hydrochloride)	11.45	0.00	1.20	down				
		Ambolic acid					10.63	0.00	1.15	up
Nucleosides, nucleotides and analogues (3)		Uridine 5′-Monophosphate Disodium Salt					14.09	0.00	1.15	down
	Purine nucleosides	8-Hydroxyguanosine	10.38	0.02	1.16	down				
	Pyrimidine nucleotides	Uridine 5′-Monophosphate	13.62	0.00	1.20	down				
Organic acids and derivatives (6)	Carboxylic acids and derivatives	*D*-Ornithine (Hydrochloride)	15.27	0.00	1.20	down				
		L-Phenylalanyl-L-Tryptophan (Phenylalanyltryptophan)	12.06	0.02	1.16	down				
		H-D-Cis-Hyp-OH					15.04	0.01	1.14	down
		Triglochinic Acid					12.06	0.00	1.16	down
		Cinnamoylglycine					10.55	0.00	1.16	down
	Hydroxy acids and derivatives	3-Hydroxyglutaric Acid	13.91	0.00	1.20	up				
Organic oxygen compounds (11)	Organooxygen compounds	Quininic Acid	12.24	0.03	1.14	up	12.77	0.00	1.15	up
		Nicotinamide Riboside (Chloride)	20.58	0.02	1.15	down	18.02	0.04	1.08	down
		Cornuside	12.04	0.00	1.20	down	14.94	0.01	1.14	down
		Sequoyitol	17.14	0.00	1.20	down				
		Acetylpyrazine					16.11	0.00	1.16	up
		4-Hydroxyacetophenone					15.66	0.00	1.16	up
		Fructose					11.82	0.03	1.09	up
		Inositol					18.63	0.02	1.12	down
		Primin-1					14.43	0.01	1.14	down
		3-Hydroxy-4-Methoxyacetophenone					14.29	0.00	1.15	down
		Styrene-Cis-2,3-Dihydrodiol	12.73	0.00	1.19	down				
Organoheterocyclic compounds (2)	Pyridines and derivatives	Nicotinamide	14.43	0.00	1.20	up	15.78	0.00	1.16	up
	Quinolines and derivatives	(2*S*,3*S*)-2-(3,4,5-trihydroxyphenyl)-3,4-dihydro-2*H*-chromene-3,5,7-triol					10.34	0.00	1.16	down
Phenylpropanoids and polyketides (36)	2-Arylbenzofuran flavonoids	Moracin C	12.14	0.01	1.18	down				
	Cinnamic acids and derivatives	(*E*)-M-Coumaric Acid	16.70	0.00	1.20	up	17.01	0.00	1.15	up
		*p*-Coumaryl Alcohol	11.37	0.02	1.16	up	14.24	0.00	1.16	up
		Ferulamide	17.32	0.00	1.19	up				
		(*E*)-Ferulic Acid	19.47	0.00	1.20	down				
		Calceolarioside B					13.67	0.03	1.10	down
		*p*-Hydroxyphenethyl Trans-Ferulate					10.64	0.00	1.15	down
	Coumarins and derivatives	Cichoriin	12.72	0.00	1.19	down				
		Norbraylin					14.61	0.00	1.15	up
		Herniarin					14.55	0.00	1.16	up
		5,7,8-Trimethoxycoumarin					15.17	0.03	1.09	down
	Flavonoids	Magnolioside	13.82	0.00	1.20	up	16.84	0.00	1.16	up
		Aristolone-2	16.49	0.01	1.17	up	16.80	0.01	1.13	up
		Tricetin	17.65	0.01	1.19	down	18.88	0.00	1.15	down
		Herbacetin	16.70	0.03	1.15	down	16.85	0.01	1.15	down
		(+)-Gallocatechin	14.66	0.00	1.20	down	16.57	0.00	1.16	down
		5-Hydroxy-7,8-Dimethoxy (2*R*)-Flavanone-5-*O*-β-*D*-Glucopyranoside	13.90	0.00	1.20	down	13.06	0.00	1.15	down
		Isoquercetin	20.58	0.02	1.16	up				
		5,7-Dimethoxyluteolin	16.40	0.00	1.20	up				
		Neoisoastilbin	24.03	0.00	1.20	down				
		Quercetagitrin	20.24	0.01	1.17	down				
		(+)-Taxifolin	17.20	0.00	1.20	down				
		Taxifolin	17.01	0.00	1.20	down				
		Quercetin 3-*O*-Neohesperidoside	15.51	0.00	1.20	down				
		Velutin	13.82	0.01	1.19	down				
		Vicenin 2	12.47	0.01	1.18	down				
		Methyllinderone	11.55	0.00	1.20	down				
		2-Phenylethylamine	10.62	0.00	1.20	down				
		(−)-Epigallocatechin					16.37	0.00	1.15	down
		Cyanidin 3-Rutinoside					15.88	0.02	1.11	down
		Methylnissolin-3-*O*-Glucoside					14.49	0.00	1.16	down
		Tiliroside					13.98	0.01	1.13	down
		Theaflavin-3-Gallate					11.74	0.01	1.13	down
	Isoflavonoids	Genistein	10.69	0.01	1.17	up				
	Linear 1,3-diarylpropanoids	Loureirin C					12.21	0.00	1.16	down
	Phenylpropanoic acids	(−)-Catechin Gallate	19.60	0.01	1.17	down	22.03	0.00	1.16	down
Unknown (2)		2-Hydroxy-3-Methylbenzalpyruvate					14.97	0.01	1.14	up
		Nudifloside D					10.27	0.01	1.13	down

Note: VIP, variable importance in projection. FC (ABS), absolute value of log2fold change.

**Table 2 metabolites-14-00484-t002:** Comparative regulation (up/down) of the mutually shared and unique metabolites between the HQI/LFI and HQO/LFO groups.

Superclass	Class (HMDB_Taxonomy)	Metabolite	HQI/LFI				HQO/LFO			
			log2FC (ABS)	*p*-Value	VIP	Regulated	log2FC (ABS)	*p*-Value	VIP	Regulated
Alkaloids and derivatives (2)		Acetyl-*L*-Carnitine (Hydrochloride)	11.71	0.00	1.24	down				
		Salsolidine					12.21	0.04	1.11	down
Benzenoids (5)	Benzene and substituted derivatives	4-Methoxycinnamyl Alcohol	20.72	0.00	1.23	up	21.06	0.00	1.20	up
		4-Acetoxy-3,5-Dimethoxybenzoic Acid	20.23	0.00	1.24	down				
		2,6-Dihydroxybenzoic Acid					16.09	0.00	1.20	up
		Dibenzyl Disulfide					13.90	0.00	1.20	up
	Phenols	Syringylpropane	15.94	0.02	1.20	up				
Lipids and lipid-like molecules (11)	Fatty acyls	Suberic Acid	20.54	0.00	1.24	down				
		Dodecanedioic Acid					14.58	0.01	1.17	down
	Glycerophospholipids	(2*S*,3*R*,4*S*)-4-Hydroxyisoleucine	12.66	0.01	1.22	up				
	Prenol lipids	(*R*)-(+)-Citronellal	15.01	0.00	1.24	down				
		Isopimaric Acid	13.86	0.00	1.24	down				
		*α*-Boswellic Acid					13.39	0.00	1.20	up
		Carvacrol					12.06	0.00	1.20	up
		Monotropein					11.58	0.01	1.19	up
	Steroids and steroid derivatives	Isomangiferolic Acid	14.26	0.00	1.24	up	12.35	0.00	1.20	up
		Estradiol	10.55	0.00	1.24	down				
		2-Phenylethylamine (Hydrochloride)					13.01	0.00	1.20	up
Nucleosides, nucleotides and analogues (2)	Pyrimidine nucleotides	Uridine 5′-Monophosphate					14.64	0.00	1.20	up
		Uridine 5′-Monophosphate Disodium Salt					13.37	0.00	1.20	down
Organic acids and derivatives (6)	Carboxylic acids and derivatives	*D*-Ornithine (Hydrochloride)					12.62	0.03	1.13	up
		*L*-Phenylalanyl-*L*-Tryptophan					12.19	0.00	1.20	up
		H-D-*Cis*-Hyp-OH					13.50	0.00	1.19	down
		Triglochinic Acid					13.06	0.00	1.20	down
		Cinnamoylglycine					12.44	0.01	1.19	down
	Hydroxy acids and derivatives	3-Hydroxyglutaric Acid	14.26	0.00	1.24	up				
Organic nitrogen compounds (1)	Organonitrogen compounds	*D*-Erythro-Sphingosine					15.27	0.00	1.20	up
Organic oxygen compounds (7)	Organooxygen compounds	Sequoyitol	17.14	0.00	1.23	down				
		Acetylpyrazine	12.07	0.00	1.24	down				
		4-Hydroxyacetophenone	11.81	0.00	1.24	down				
		Inositol					15.95	0.02	1.17	down
		3-Hydroxy-4-Methoxyacetophenone					12.83	0.00	1.19	down
		Primin-1					12.64	0.00	1.20	down
		Styrene-*Cis*-2,3-Dihydrodiol					11.77	0.02	1.16	up
Organoheterocyclic compounds (1)	Indoles and derivatives	2-(4-Hydroxy-2-Oxoindolin-3-Yl) Acetonitrile	14.24	0.00	1.23	down	14.36	0.00	1.20	down
Phenylpropanoids and polyketides (29)	Cinnamic acids and derivatives	Ferulamide	10.37	0.00	1.24	up				
		(*E*)-Ferulic Acid					15.83	0.01	1.19	up
		Calceolarioside B					13.34	0.01	1.18	down
	Coumarins and derivatives	Herniarin	14.96	0.02	1.19	down				
		Norbraylin	13.78	0.00	1.23	down				
		Cichoriin	12.72	0.00	1.23	down				
		5,7,8-Trimethoxycoumarin					11.02	0.01	1.19	down
	Flavonoids	5-Hydroxy-7,8-Dimethoxyflavanone	21.19	0.00	1.24	down	22.04	0.00	1.20	down
		Farrerol	11.35	0.01	1.22	down	18.44	0.00	1.20	down
		Padmatin	15.79	0.00	1.23	down	16.45	0.00	1.20	down
		4′-Hydroxy-5,7-Dimethoxyflavanone	12.15	0.00	1.24	down	13.54	0.00	1.20	down
		Gossypin	15.01	0.03	1.17	down	11.90	0.00	1.20	down
		Norwogonin	12.55	0.00	1.24	down	11.75	0.03	1.13	down
		Isoquercetin	21.72	0.01	1.21	up				
		(−)-Epigallocatechin	16.37	0.00	1.23	up				
		Neoisoastilbin					23.78	0.00	1.20	up
		(+)-Taxifolin					19.75	0.00	1.20	up
		Taxifolin					19.44	0.01	1.19	up
		Velutin					16.52	0.00	1.20	up
		2-Phenylethylamine					12.49	0.05	1.10	up
		Methyllinderone					11.54	0.00	1.20	up
		Methylnissolin-3-*O*-Glucoside					17.69	0.01	1.18	down
		Tiliroside					16.82	0.00	1.20	down
		5,7-Dimethoxyluteolin					16.40	0.00	1.20	down
		Cyanidin 3-Rutinoside					12.76	0.01	1.18	down
	Isoflavonoids	Genistein	10.62	0.00	1.24	up				
	Phenylpropanoic acids	Piscidic Acid					10.40	0.00	1.20	up
	Tannins	3,8-Di-*O*-Methylellagic Acid	18.34	0.00	1.24	up	16.22	0.00	1.20	up
	Linear 1,3-diarylpropanoids	Loureirin C					11.77	0.00	1.20	down
Unknown (2)		2-Hydroxy-3-Methylbenzalpyruvate	14.69	0.00	1.24	down				
		H-Gly-Pro-OH					12.51	0.00	1.19	up

Note: VIP, variable importance in projection. FC (ABS), absolute value of log2 fold change.

**Table 3 metabolites-14-00484-t003:** The significantly regulated metabolites in the network analysis in the different groups.

Metabolite ID	Metabolite	KEGG_Annotation	KEGG_Pathway_Annotation
HQI/HQO			
POS_q83	Apiin	C04858	Flavone and flavonol biosynthesis (ko00944)
NEG_q313	Myricetin	C10107	Flavonoid biosynthesis (ko00941) Flavone and flavonol biosynthesis (ko00944) Biosynthesis of secondary metabolites (ko01110)
POS_t44	5-Hydroxyisourate	C11821	Purine metabolism (ko00230) Metabolic pathways (ko01100)
NEG_t83	*S*-(4-Methylthiobutylthiohydroximoyl)-*L*-cysteine	C17242	Glucosinolate biosynthesis (ko00966) Biosynthesis of secondary metabolites (ko01110) 2-Oxocarboxylic acid metabolism (ko01210)
POS_q213	*L*-Arginine	C00062	Arginine biosynthesis (ko00220) Monobactam biosynthesis (ko00261) Arginine and proline metabolism (ko00330) D-Amino acid metabolism (ko00470) Aminoacyl-tRNA biosynthesis (ko00970) Metabolic pathways (ko01100) Biosynthesis of secondary metabolites (ko01110) Biosynthesis of amino acids (ko01230) ABC transporters (ko02010)
LFI/LFO			
NEG_q424	Tricetin	C10192	Flavonoid biosynthesis (ko00941)
NEG_q207	(+)-Gallocatechin	C12127	Flavonoid biosynthesis (ko00941) Biosynthesis of secondary metabolites (ko01110)
POS_q148	(−)-Epigallocatechin	C12136	Flavonoid biosynthesis (ko00941) Biosynthesis of secondary metabolites (ko01110)
NEG_q160	Dihydromyricetin	C02906	Flavonoid biosynthesis (ko00941) Biosynthesis of secondary metabolites (ko01110)
NEG_q313	Myricetin	C10107	Flavonoid biosynthesis (ko00941) Flavone and flavonol biosynthesis (ko00944) Biosynthesis of secondary metabolites (ko01110)
HQI/LFI			
POS_q148	(−)-Epigallocatechin	C12136	Flavonoid biosynthesis (ko00941) Biosynthesis of secondary metabolites (ko01110)
HQO/LFO			
NEG_q431	Uridine 5′-Monophosphate	C00105	Pyrimidine metabolism (ko00240) Metabolic pathways (ko01100) Nucleotide metabolism (ko01232) Biosynthesis of cofactors (ko01240)

Note: “Metabolite (ID)” refers to the specific metabolite identified; the numbers in parentheses are the KO classification.

## Data Availability

The data presented in this study are available on request from the corresponding author. The data are not publicly available due to privacy.

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
