# Peer review of "Drug or Toxic? A Brief Understanding of the Edible Corolla of Rhododendron decorum Franch. by Bai Nationality with Comparative Metabolomics Analysis"

_metabolites, 2024, doi:10.3390/metabo14090484_

Round 1
Reviewer 1 Report
Comments and Suggestions for Authors
The manuscript is devoted to an interesting and popular area of ​​research - the study of low-molecular metabolites of renewable plant materials. The presented work deals with the species Rhododendron decorum, or more precisely, the inflorescences of this plant, which are included in the diet. In my opinion, there are two important aspects of this manuscript that need to be clarified. This will allow to evaluate the correctness of the results and discussions presented.
1. Method of analysis. The authors chose mass spectrometry, which is dominant in this type of studies. However, the choice of an instrument with linear ion trap (and/or triple quadrupole) for non-targeted screening of hundreds of metabolites is not justified. This is a low resolution, there are no standard samples (based on the experimental part), this casts doubt on the correctness of the identification. In describing the detection conditions, the authors refer to a study on serum using high-resolution mass spectrometry. Will the analytes be the same? The use of metabolite elution times (from the cited work) under different chromatographic conditions (column, flow rate) is also questionable. Authors are required to revise the manuscript in such a way as to eliminate this doubts.
2. If the authors will demonstrate that the component composition of the extracts has been correctly determined, they should also clarify the results of the chemometric methods. For example, Figure 2. In essence, the authors compare two objects (two parts of the plant). Each of them has 3 samples (3 extractions, which are essentially parallel analyses). Is it correct to use such chemometrics and draw any conclusions only by comparing the pair with each other. Principal component analysis is used for large data sets and allows finding certain patterns and differences. This approach will always find large differences if only two samples are examined.
Based on the above, authors are required to conduct a major revision of their manuscript and eliminate possible inconsistencies.
Reviewer 2 Report
Comments and Suggestions for Authors
The article "Drug & Toxic? A Brief Understanding of the Edible Corolla of Rhododendron decorum Franch. by Bai Nationality with Comparative Metabolomics Analysis" can be published after revision.
-in Table 1 and Table 2:
- R and S in (2S,3R,4S)-4-Hydroxyisoleucine or (R)-(+)-Citronellal...and so on... must be italics
- cis in Styrene-Cis-2,3-Dihydrodiol and Styrene-Cis-2,3-Dihydrodiol must be italics cis
- S and H in (2S,3S)-2-(3,4,5-trihydroxyphenyl)-3,4-dihydro-2H-chromene-3,5,7-triol must be italics
- P-Coumaryl Alcohol must be p-Coumaryl Alcohol
- (E)-Ferulic Acid must be (E)-Ferulic Acid
- O-Beta-D-Glucopyranoside must be O-beta-D-Glucopyranoside or O-β-D-Glucopyranoside
- H-D-Cis-Hyp-Oh must be H-D-Cis-Hyp-OH
- The conclusions must be improved with data from the exposed research
- The bibliography must be written according to the requirements of the journal
Comments on the Quality of English Language
The English language is good.
Reviewer 3 Report
Comments and Suggestions for Authors
The current manuscript describes the importance of R. decorum in term of medicinal property. Here authors showed significant differences in metabolites profiles in different parts mainly floral organs. Authors also did metabolomics with relation to the geographical regions. On the basis of their results authors claims that R. decorum Franch. populations in Heqing county exhibited higher accumulation of secondary metabolites within their flowers rendering them more advantageous for medicinal purposes albeit potentially more toxic. The finding is very interesting, but I have few concern regarding their claims.
1. Authors did not perform any toxicity experiment.
2. They did not check any gene expression with respects to the metabolites.
Without these experiment it is not possible on the definite conclusions. If authors perform these experiment the manuscript improve significantly.
Comments on the Quality of English LanguageMinor spell check is required
Round 2
Reviewer 1 Report
Comments and Suggestions for Authors
The authors revised the manuscript, taking into account both key issues for this work. In general, the work appears to be a logically structured and complete study. There are no more significant additions or comments in the work.